# Characteristics of Hydroxyapatite-Modified Coatings Based on TiO_2_ Obtained by Plasma Electrolytic Oxidation and Electrophoretic Deposition

**DOI:** 10.3390/ma16041410

**Published:** 2023-02-08

**Authors:** Roxana Muntean, Mihai Brîndușoiu, Dragoș Buzdugan, Nicoleta Sorina Nemeș, Andrea Kellenberger, Ion Dragoș Uțu

**Affiliations:** 1Department of Materials and Manufacturing Engineering, Politehnica University Timișoara, Piața Victoriei 2, 300006 Timișoara, Romania; 2Research Institute for Renewable Energy—ICER, Politehnica University Timișoara, Piața Victoriei 2, 300006 Timișoara, Romania; 3Faculty of Industrial Chemistry and Environmental Engineering, Politehnica University Timișoara, Piața Victoriei No. 2, 300006 Timișoara, Romania

**Keywords:** plasma electrolytic oxidation, titanium, hydroxyapatite, corrosion and wear resistance

## Abstract

In order to modify the surface of light metals and alloys, plasma electrolytic oxidation (PEO) is a useful electrochemical technique. During the oxidation process, by applying a positive high voltage greater than the dielectric breakdown value of the oxide layer, the formation of a ceramic film onto the substrate material is enabled. The resulting surface presents hardness, chemical stability, biocompatibility, and increased corrosion wear resistance. The current study aims to investigate the corrosion resistance and tribological properties of PEO-modified coatings on titanium substrates produced by applying either direct or pulsed current in a silicate-alkaline electrolyte. In this way, a uniform TiO_2_ layer is formed, and subsequently, electrophoretic deposition of hydroxyapatite particles (HAP) is performed. The morpho-structural characteristics and chemical composition of the resulting coatings are investigated using scanning electron microscopy combined with energy dispersive spectroscopy analysis and X-ray diffraction. Dry sliding wear testing of the TiO_2_ and HAP-modified TiO_2_ coatings were carried out using a ball-on-disc configuration, while the corrosion resistance was electrochemically evaluated at 37 °C in a Ringer’s solution. The corrosion rates of the investigated samples decreased significantly, up to two orders of magnitude, when the PEO treatment was applied, while the wear rate was 50% lower compared to the untreated titanium substrate.

## 1. Introduction

The outstanding properties of titanium and its alloys (high strength-to-weight ratio, superior corrosion resistance, or biocompatibility) make them suitable for various advanced engineering applications, such as aerospace, marine, photocatalysis, or biomedical [1,2,3]. However, there are still various problems associated with the use of such materials in terms of surface features to ensure good stability, wear resistance, or bonding strength [2]. Although titanium performs well in most corrosive environments, it can be susceptible to corrosion under highly aggressive conditions. Commonly, titanium is very reactive in its metallic state and naturally forms a thin stable oxide layer (passivation layer typically between 1.5–5 nm) when exposed to normal conditions. Still, once high contact loads are applied combined with reducing or low oxygen environments, the native oxide film may be rapidly damaged, promoting galvanic or crevice corrosion. Consequently, several superficial treatment processes are frequently applied to enhance surface characteristics [3,4], including physical vapor deposition, thermal oxidation, plasma spraying, anodizing, or plasma electrolytic oxidation (PEO) [5]. Moreover, the tribological behavior of Ti-based materials can be significantly improved by applying an appropriate surface modification strategy [6]. The PEO is a widely used superficial treatment process to shield the surfaces of light metals and alloys by inducing the growth of an inert oxide layer. Since its first application in the early 1990s [3], scientists have constantly attempted to understand and improve this technology by varying the main process parameters and searching for new possible applications. The PEO process consists of a luminescent phenomenon of electrical discharge on the metallic working part (immersed in an electrolyte) that acts similar to an anode, accompanied by many crystallizing and melting events [7]. The oxidation occurs when a positive voltage greater than the dielectric breakdown value of the oxide film is reached between the sample and the counter electrode [8], and in this way, a highly adherent ceramic-like porous layer is formed. The coating consists of two different layers, the outer one, which presents numerous structural defects (pores and cracks), and the inner layer, which is denser and more compact [8]. Due to the thermochemical interactions and sparking discharges between the electrolyte and the metallic substrate, a localized high pressure (10^2^–10^3^ MPa) and temperature (10^3^–10^4^ K) are achieved [9]. Adjusting the current density and mode, voltage, or power provided to the cell, a wide range of polarization conditions, including direct current (DC) [10], alternating current (AC) [11], or pulsed current (PC) [12], are accessible for the production of such coatings. The PEO process has several advantages compared to other techniques, such as cost-effectiveness, eco-friendly, rapid and easy conversion of the metal surface [13], facile control of coating thickness and porosity, and finely tailored coatings, considering their chemical composition and microstructure [14]. Furthermore, the electrolyte used during the PEO process can strongly define the properties of the treated surfaces, including the microstructure or crystallinity [8,9,15]. PEO-developed coatings on titanium substrates exhibit great importance, primarily in medical applications, due to their ductility, biocompatibility, and corrosion resistance. In this case, to improve osteointegration, the incorporation of elements such as calcium, phosphorous, or silicon is frequently reported [16,17]. Biofunctionalization consists mainly of two different approaches: either the incorporation of inorganic components and their further conversion during the plasma treatment or the integration of organic substances directly into the oxide pores [18]. Consequently, electrolytes containing calcium acetate and glycerophosphate [2,8,14,16], sodium silicate and hexametaphosphate [4,13,15], phosphoric acid and copper nitrate [10,11], calcium citrate and potassium titanium oxalate [17], sodium aluminate and potassium dihydrogen phosphate [19] or molten salts mixtures (KNO_3_ and NaNO_3_) [12] are commonly used. It has been shown that PEO-modified titanium from a silicate-containing solution presents better osteointegration compared to untreated titanium [20]. Several studies in the literature confirm the significant effect of the electrolyte composition on the corrosion behavior of the PEO-developed coatings [9,13]. Shokouhfar et al. [13] concluded that changing the electrolyte composition without altering the rest of the process parameters has led to an increase in pores size generated on the surface of the samples, and furthermore, the corrosion resistance was considerably affected. Molaei et al. [9] investigated the effect of different sodium-based additives in the working electrolyte on the corrosion resistance of the PEO-modified pure titanium. Their study revealed that the additives have a clear influence on the microstructure, phase composition, and subsequently on the corrosion resistance of the PEO-modified samples. Another approach frequently reported in the literature is the production of composite PEO coatings by adding different particles, either directly in the working electrolyte during the PEO treatment or as an additional step once the oxidation process is completed [8]. In this way, several characteristics can be acquired, such as an antifouling effect, better adhesion, hardness, wettability, or better biological response.

The aim of the current study is to assess the corrosion resistance and wear behavior of hydroxyapatite-modified TiO_2_ coatings produced by a two-step fabrication method. Firstly, a PEO treatment is proposed based on two different polarization conditions (either direct current or pulsed current) from a sodium silicate and potassium hydroxide alkaline solution, and subsequently, uniform incorporation of hydroxyapatite particles (HAP) is achieved by electrophoresis from an additive-free HAP suspension in isopropanol. 

## 2. Materials and Methods

Commercially available pure titanium sheets (grade 1) of 3.5 mm thickness were cut in the form of rectangular samples with the dimensions of 50 mm × 20 mm and used further as substrate material. Prior to the PEO, the samples were ground with SiC paper (up to 4000 grit) and washed in distilled water and ethanol for 5 min in an ultrasonic bath. The surface roughness of the substrate was approximately 0.15 ± 0.02 μm. The PEO process was carried out in a single-compartment two-electrode cell containing 500 mL alkaline electrolyte prepared by mixing 20 g L^−1^ Na_2_SiO_3_ with 4 g L^−^^1^ KOH (for increasing the electrical conductivity), in distilled water, at a pH of 12. The solution was continuously stirred during the oxidation process using a magnetic stirrer to keep the temperature as low as possible (under 40 °C) and to avoid concentration gradients. The working electrodes (WE) in the form of Ti samples (employed as anodes) were immersed in the electrolyte using a threaded rod that assured electrical contact, and for the counter electrode (CE) (cathode), a rectangular stainless-steel plate, 70 mm × 100 mm (type 304) was employed. Two different polarization conditions were applied with the aid of an electrical power supply (max. 250 V/20 A, RFT Sparstelltrafo SST 250/20), one using direct current (DC) and the second one with pulsed current (PC). The process was carried out in galvanostatic mode at a total current value of 200 mA for all the samples, while the maximum reached voltage was 240 V. In the case of PC treatment, a unipolar current was applied with the aid of an electrical relay, setting a duty cycle (*d.c*) of 20%, at a frequency of 0.2 Hz. Subsequently, the PEO-treated samples were washed with distilled water and dried under warm air. From the prepared samples, the Ti PEO PC samples were selected and further subjected to the electrophoretic deposition of HAP from an additive-free suspension (2.5% wt.) prepared using only hydroxyapatite particles (commercially available p.a. ≥ 90%, Ca_3_(PO_4_)_2_, Fluka Analytical) and isopropanol (99.9%). The commonly used chemicals reported in the literature for the preparation of HAP suspensions encompass the HAP particles, a solvent (n-butanol, isopropanol), a dispersant (triethanolamine, iodine), and sometimes other additives, such as PVP (polyvinyl pyrrolidone), PEG (polyethylene glycol), PEI (polyethylene amine). However, such additives are not desirable since, after the drying step, depending on their nature, they may leave residues on the surface of the modified sample. In this case, a graphite plate (70 mm × 100 mm × 5 mm) was used as an anode (CE), and the working electrode consisted of previously modified Ti-PEO samples. The schematic representation of the complete PEO process is shown in Figure 1, and the PEO treatment conditions are presented in Table 1.

The surface morphology of the resulting samples was investigated using an FEI Quanta™ FEG 250 (FEI Quanta™, Hillsboro, OR, USA) scanning electron microscope (SEM) equipped with an energy-dispersive spectroscopy analyzer (EDS) for chemical composition identification. For structure and phase evaluation, an X-ray Diffractometer (XRD) PANalytical X’Pert Pro Powder (Malvern Panalytical, Malvern, UK) was employed, with Cu-*K_α_* radiation (*λ* = 1.54), and the measurements were carried out between 20° and 100° 2ϴ-angles, with a step size of 0.033, at 45 kV and 30 mA. The roughness of the samples before and after the PEO treatment was evaluated with a Mitutoyo SJ-201 portable surface tester (Mitutoyo, Aurora, IL, USA). The thicknesses of the coatings were checked with an eddy current thickness gauge (Namicon, Otopeni, Romania).

The corrosion resistance of PEO-treated samples and HAP-modified coatings was investigated via open circuit potential (OCP), potentiodynamic polarization, and electrochemical impedance spectroscopy (EIS) measurements and compared to the Ti substrate. All electrochemical measurements were carried out in a water jacket three-electrode configuration corrosion cell connected to an Autolab PGSTAT 302 N potentiostat/galvanostat (Metrohm, Herisau, Switzerland). The working electrode consisted of a titanium sample (1 cm^2^ active surface) modified by PEO as previously described, the reference was an Ag/AgCl (3M NaCl) electrode positioned near the working electrode via a Luggin capillary, and the counter electrode was a Pt gauze. Potentiodynamic polarization curves were recorded in a Ringer’s solution at 37 °C with a scan rate of 0.16 mV s^−1^. The Ringer’s solution was prepared by mixing 9 g L^−1^ NaCl, 0.42 g L^−1^ KCl, 0.2 g L^−1^ NaHCO_3_, and 0.24 g L^−1^ CaCl_2_ with distilled water. The main corrosion parameters were evaluated from the polarisation curves using the Tafel extrapolation method in the linear region of the anodic and cathodic branches. EIS measurements were carried out in the same conditions and electrolyte, at the corrosion potential value, by collecting 60 points for each spectrum, in the frequency range from 10^5^ to 10^−2^ Hz and AC voltage amplitude of 10 mV rms. An electrical equivalent circuit was proposed to model the experimental EIS data by a complex non-linear least squares procedure using the ZView 3.0 software (Scribner Associates, Inc., Southern Pines, NC, USA). The dry sliding wear behavior of the Ti substrate and PEO-modified samples was evaluated using the ball-on-disc method, according to the ASTM standard G99. The tests were performed with the aid of a TR-20 Tribometer, Ducom-Materials Characterisation Systems, under dry sliding conditions and ambient temperature, with a normal load of 5 N, sliding rate of 300 rot min^−1^, test time *t* = 100 min, and a total sliding distance around 1000 m. The static partner selected for the tests was a 100Cr6 steel ball of 6 mm diameter. During testing, the variation of the coefficient of friction (COF) with the distance was automatically recorded. The profile of the wear track was subsequently determined using a 3D optical profilometer, (Ducom-Materials Characterization Systems, Groningen, The Netherlands), and the loss of material and wear rate were estimated. For each sample, three measurements were performed in order to ensure the reproducibility of the results.

## 3. Results and Discussion

### 3.1. Coating Morphology and Composition

The PEO treatment changed the surface of the titanium samples from bright metallic to a porous dark grey ceramic coating, typical for a PEO-modified surface. Macroscopically, the samples have the same appearance, but the microscopic analysis revealed some significant differences in the microstructure of the samples due to the different treatment conditions. The SEM micrographs from Figure 2 show the complex porous surface of the oxidized titanium samples in different stages. According to the PEO process mechanism described in the literature [5], initially, at low voltage values, a uniform, dense, and thin (nano-scaled) oxide layer is formed on the surface of the samples. As soon as the applied voltage reaches the breakdown potential of the previously formed film, various sparking phenomena appear, leading to the formation of discharge channels, where temperature and pressure achieve higher values and different processes, such as melting and oxidation, may occur. During the PEO process, molten oxides may migrate through the discharge channels and then are rapidly cooled, generating an uneven microstructure with well-separated micropores, similar to a volcano-crater shape [9]. When DC is applied, the oxidized titanium surface presents numerous irregularities and open pores, which may be attributed to the intense gas evolution that takes place during the oxidation process. It is considered that oxygen is generated at the limit of the crystals [4], and the gas is released when the oxide film breaks, producing well-defined pores. Moreover, during the DC discharges, the heating process is more intense and may melt the oxides and eject them from the coating/substrate interface to the surface of the coating, promoting a higher roughness [21]. In these conditions, the surface roughness increases from the initial titanium substrate value *Ra* = 0.15 μm to *Ra* = 0.4 μm, as shown in Figure 3. In contrast, the pulsed current treatment generates a smoother surface, with smaller pores and a denser structure, the roughness reaching, in this case, almost 0.3 μm, confirming the microstructure of the PEO-modified surfaces is strongly influenced by the polarization conditions [22]. This aspect may be attributed to the intermittent (pulsed) current applied during the oxidation (*t_on_* = 1 s and *t_off_* = 4 s, for 250 cycles), the pauses between pulses providing enough time to avoid the overheating of the surface, enabling better control of the intensity of the micro-discharges and delivering smoother coatings, as the gas evolution is also limited [22]. Generally, the PEO treatment regime and duration directly influence the thickness and the roughness of the PEO-produced coatings. In fact, the applied potential is the main factor controlling these characteristics. In the DC regime, in the range of low potentials, up to 150 V, the thickness increases slightly at the beginning of the oxidation, and after the breakdown potential is reached, the coating growth is accelerated. In this case, the Ti PEO DC coatings reach an average value of 35 μm. In the PC treatment, where it is possible to control the duration and intensity of the discharges by adjusting the *t_on_*, *t_off_*, and *d.c.*, the thickness of the TiO_2_ coatings reached only 12–16 μm, taking into account that the total oxidation time is also reduced compared to DC regime. 

Based on the EDS spectra (Figure 4a,b), it was found that both PEO-produced coatings consist of titanium, oxygen, and silicon, the last element arriving from the chemical composition of the electrolyte. For the HAP electrophoretic deposition, the Ti PEO PC-modified samples were selected, as they provide a surface with fewer defects compared to the DC-treated ones. The morphology of the HAP-modified sample (Figure 2c) indicates the presence of fine HAP uniformly distributed onto the surface of the oxide coating in brighter areas, filling the majority of gaps and pores. In this way, the surface roughness is reduced to approx. 0.2 μm. The EDS spectrum (Figure 4c) confirms the presence of HAP particles since the characteristic peaks for calcium and phosphorous are identified in addition to the ones for Ti and O, increasing in this way the biocompatibility degree of the modified surfaces [23].

Figure 5 displays the representative diffraction patterns of the PEO-modified samples, including the one obtained for pure titanium substrate, as a reference. The untreated Ti is entirely formed of hexagonal *α*-phase (denoted Titanium in Figure 5), with the preferred orientation of the crystallites in (002), (101), and (013) directions. Since the XRD analysis detects the full-thickness coating, also reaching the substrate, indicating that the thickness of the coatings is not very high, the main characteristic peaks for titanium are also visible in the PEO-modified samples’ patterns. A change in the intensity of the Ti (002) reflection was observed in the case of PEO-treated samples, and in addition to the titanium characteristic peaks, the TiO_2_ phase, in the form of anatase, is detected for all treated samples. The dominant reflection for anatase is identified at 2*ϴ* angles of 25° (101). This phase typically forms during the PEO treatment in alkaline solutions with high electrical conductivity as a reaction between Ti^4+^ and OH^−^ due to the high pressure and temperature reached in the discharge channels. It has been previously reported that the applied voltage value directly influences the crystallinity degree of the titanium surfaces [24]. Consequently, there are no significant differences between the PEO-treated samples regarding the XRD patterns since all of the samples were prepared using similar voltage values in the same electrolyte. The peaks corresponding to HAP were not identified in the XRD patterns, as the weight percent of hydroxyapatite on the PEO-treated samples is under the detection limit of this method. Additional phases derived from the electrolyte chemical composition (such as Si-containing phases) may be found in the coating, either in an amorphous or crystalline structure, but the content is below the detection limit of XRD [25].

### 3.2. Corrosion Behavior

The corrosion resistance of the titanium-modified samples and titanium substrate used as reference was assessed with OCP measurements, potentiodynamic polarization, and EIS tests. The OCP variation during 600 s in Ringer’s solution at 37 °C can be found in Figure 6. The titanium substrate stabilizes at less noble potential compared to the PEO-treated samples. The most promising behavior in terms of corrosion performance can be observed for the Ti PEO DC, where higher OCP values are identified. The potentiodynamic polarization curves, in the semilogarithmic form (Figure 7), deliver important information regarding the corrosion behavior of the titanium-modified surfaces. Significant differences in both corrosion current density and potential can be observed between the PEO-modified samples and titanium substrate. The anodic branch of the polarization curves was shifted to lower current densities after the PEO treatment, compared to the Ti substrate, denoting higher stability. In fact, analyzing the values presented in Table 2, an increase in the corrosion potential of about 0.23 V is observed for the Ti PEO DC compared to the Ti substrate, while the corrosion current density decreases up to two orders of magnitude due to the presence of a compact TiO_2_ layer formed during the PEO process. Similar behavior was also observed for the PC-treated samples. In the case of HAP-modified titanium, even if the corrosion potential presents a more negative value, the addition of HAP does not alter the corrosion performance, and the current density remains low, comparable with the PEO-treated samples. Furthermore, the corrosion rates decreased significantly for the samples treated with PEO. Although the Ti PEO DC sample presents larger micro-pores and roughness compared to the other modified samples, which should increase the probability for the adsorption of Cl^−^ anions on the surface, the better corrosion behavior may be attributed to the thicker oxide layer previously formed in the PEO process. Based on the potentiodynamic polarization curves, the corrosion resistance increases in the order of Ti substrate < Ti PEO PC + HAP < Ti PEO PC ≈ Ti PEO DC. To investigate more deeply the corrosion mechanism of the titanium-based samples, EIS measurements were performed in the same electrolyte and similar testing conditions. Figure 8 gives the results of impedance measurements represented as Nyquist plots (Figure 8a,b), absolute impedance Bode plots (Figure 8c), and phase angle Bode plots (Figure 8d).

The impedance data of untreated Ti substrate indicate the presence of one semicircle in the whole frequency range (Figure 8b), corresponding to a single time constant, as indicated by a single maximum in the phase angle plot (Figure 8d). In contrast, the PEO-treated Ti samples show the appearance of a smaller radius semicircle at high frequencies, followed by a larger radius semicircle at low frequencies (Figure 8a,b), indicating the presence of two-time constants. Two maxima are also observed in the phase angle plots (Figure 8d) at low and, respectively, high frequencies. This behavior is consistent with a two-layer structure often observed in the PEO coatings [5,9,26], where the high-frequency semicircle corresponds to the outer porous layer and the low-frequency semicircle to the compact inner layer. The highest absolute impedances were observed for the Ti PEO DC and HAP-modified samples, two orders of magnitude higher than for untreated Ti and one order of magnitude higher than in the absence of HAP. This indicates similar corrosion resistance for Ti PEO DC and the samples treated using pulsed current and modified with HAP.

Figure 9 gives the two electrical equivalent circuits used to model the experimental EIS data of untreated and PEO-treated Ti samples. Both models use a constant phase element (CPE) instead of a capacitor for a better approximation of the non-ideal capacitive response of real systems. The impedance of a CPE is given by *Z_CPE_ = T*^−1^*(jω) ^−n^*, where *T* is a parameter expressed in F s*^n^*^−1^ cm^−2^, related to the capacitance, *j* is the imaginary unit, *ω* is the angular frequency in radians and *n* is an exponent between 0 and 1, describing the constant phase angle of CPE, equal to −(*n* × 90). Depending on the *n* values, a CPE behaves between the limiting cases of a resistor (if *n* = 0) or a capacitor (if *n* = 1). The model used for the untreated Ti sample (Figure 9a) has one time constant and contains the solution resistance *Rs* and a parallel connection of *R* and *CPE* corresponding to the resistance and constant phase element of the passive film on the Ti surface. The model chosen for PEO-treated samples (Figure 9b) has two time constants and includes, in addition to *Rs*, two parallel connections *R1-CPE1* and *R2-CPE2* corresponding to the resistance and capacitance of the outer porous layer and the compact inner layer, respectively. Since untreated Ti shows a single passive layer on its surface, the impedance characteristics of this layer could be assimilated to that of the inner layer in the case of PEO-treated Ti samples. 

Table 3 gives the calculated EIS parameters for untreated and PEO-modified samples, together with the experimental errors and the goodness-of-fit expressed by the Chi-squared (*χ*^2^) value. The low values of both errors and *χ*^2^ indicate good quality of fit of the experimental EIS data to the proposed electrical equivalent circuits. 

Data from Table 3 clearly show that for all PEO-treated samples, the resistance of the inner layer (*R*_2_) is higher than that of the outer layer (*R*_1_), indicating that the compact barrier layer is primarily responsible for the corrosion resistance of PEO coatings. For Ti PEO DC and Ti PEO PC samples, the resistance of the inner film is about two orders of magnitude higher than that of the outer layer, while it increases three orders of magnitude after HAP deposition. The highest resistance of the outer layer is observed for the Ti PEO DC sample (11.6 × 10^4^ Ω cm^2^) and the lowest for the Ti PEO PC sample (0.25 × 10^4^ Ω cm^2^) with a slight improvement after HAP deposition (0.40 × 10^4^ Ω cm^2^). These results are correlated with the layer thickness, as the Ti PEO DC coating exhibits the highest thickness. Based on the impedance data, it has been concluded that the corrosion resistance increases in the order of Ti substrate < Ti PEO PC < Ti PEO PC + HAP ≈ Ti PEO DC. 

### 3.3. Sliding Wear Tests

Figure 10 displays the relationship between the coefficient of friction (COF) and the sliding distance obtained during the tribological tests of the titanium substrate and PEO-modified samples versus the counter body 100Cr6 ball. It can be observed that the most important variation of COF takes place in the first 250 m of the testing distance for all the samples. After this interval, the COF values remain almost stable for the rest of the measurement. Comparing the COF variation during the tests, it can be concluded that the smallest values are achieved for the Ti substrate, as it provides the smoothest initial surface (*Ra* = 0.15 μm), leading to lower friction forces in contact. In this case, the difference between the starting COF (0.2) and the maximum reached value (0.65) denotes the poor mechanical properties, low hardness, and high wear rate of the titanium in this specific testing system due to the increase in the contact area between the sample’s surface and the counter body. In comparison, the DC and PC PEO-treated samples exhibit similar behavior, with a minimum COF value, at the beginning of the test, of around 0.5 ± 0.05 and an average COF value of around 0.8. This performance may be attributed to the initial rougher surface of the samples due to the presence of pores and grooves generated during the PEO treatment and possible detachment of particles (friability of the outer layer), but once the measurement enters the steady state, the COF remains stable. The plots presented in Figure 8 indicate that the PEO treatment leads to an increase in the COF values due to the abrasive action of the rough treated surface on the counter facing 100Cr6 ball. Even if the COF value is higher for the PEO-treated samples, the histogram presented in Figure 11 indicates lower wear rates. In both cases, the estimated wear rate is 50% lower compared to the titanium substrate. In the case of the HAP-modified sample, the COF values are significantly reduced, starting with a minimum value of 0.33 and reaching an average COF value of 0.65, similar to the Ti substrate. The deposition of HAP onto the PC PEO-treated samples resulted in a decrease in surface roughness due to the incorporation of HAP in the pores and grooves, leading to a reduced contact area between the sample and the counter body. Moreover, the wear rate is lower, compared to the titanium substrate values, as the presence of HAP leads to a more compact surface and may form, during testing, a tribofilm that reduces the contact friction between the two surfaces [27].

## 4. Conclusions

PEO coatings on high-purity titanium substrates were successfully produced from an alkaline electrolyte using two different polarization conditions. A two-step method is proposed to fabricate HAP-modified coatings. The deposition of HAP was performed by electrophoresis from an additive-free suspension. SEM micrographs revealed the formation of a TiO_2_ layer with uneven microstructure during the PEO treatment and the homogeneous deposition of HAP particles by electrophoresis. The TiO_2_ phase, in the form of anatase, is detected for all the PEO-treated samples using the XRD analysis. The corrosion resistance of the PEO coatings is significantly higher compared to untreated titanium, in the Ringer’s solution, at 37 °C. In the case of HAP-modified titanium samples, even if the corrosion potential presents a more negative value, the addition of HAP does not alter the corrosion performance, and corrosion current densities remain low, comparable to the ones obtained for PEO-treated samples. The highest absolute impedances were observed for the Ti PEO DC and HAP-modified samples, two orders of magnitude higher than for untreated Ti and one order of magnitude higher than in the absence of HAP. The coefficient of friction in the dry sliding wear tests of the coatings against a steel ball (100Cr6), using a load of 5 N, is the average value of ~0.8 compared with ~0.6 for the untreated titanium. Even if the COF values are higher for the PEO-treated samples, the estimated wear rates are 50% lower compared to the titanium substrate. The developed PEO-modified coatings may be suitable for biomedical applications.

## Figures and Tables

**Figure 1 materials-16-01410-f001:**
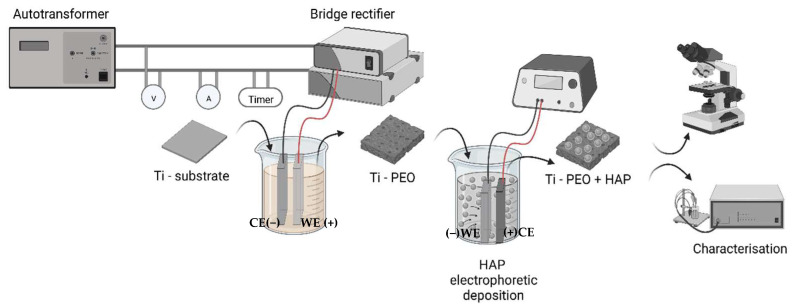
Schematic representation of the synthesis process (Created with BioRender.com–created on 12 January 2023).

**Figure 2 materials-16-01410-f002:**
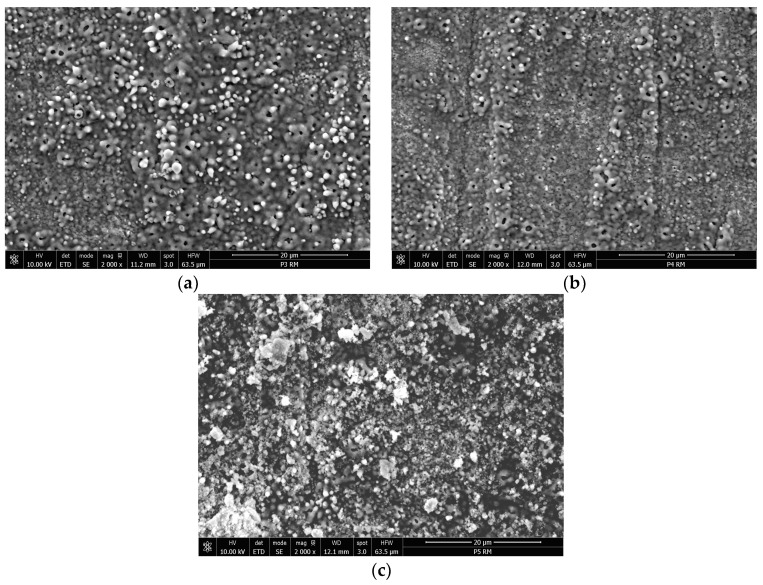
SEM micrographs of PEO-modified Ti-based samples: (**a**) Ti PEO DC; (**b**) Ti PEO PC; (**c**) Ti PEO PC + HAP.

**Figure 3 materials-16-01410-f003:**
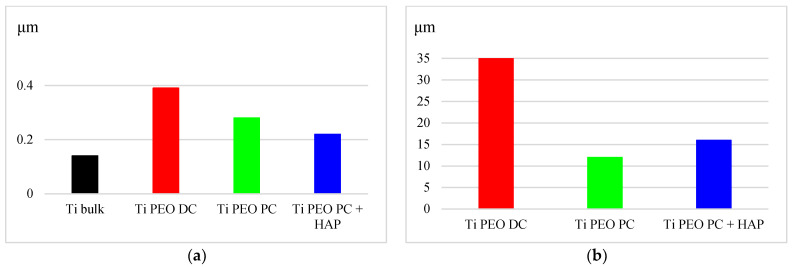
(**a**) Average surface roughness (*Ra*) of Ti substrate, Ti PEO DC; Ti PEO PC; Ti PEO PC + HAP; (**b**) Coating thickness of Ti PEO DC; Ti PEO PC; Ti PEO PC + HAP.

**Figure 4 materials-16-01410-f004:**
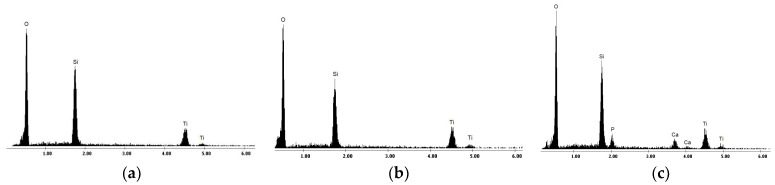
EDS spectra of Ti-based samples: (**a**) Ti PEO DC; (**b**) Ti PEO PC; (**c**) Ti PEO PC + HAP.

**Figure 5 materials-16-01410-f005:**
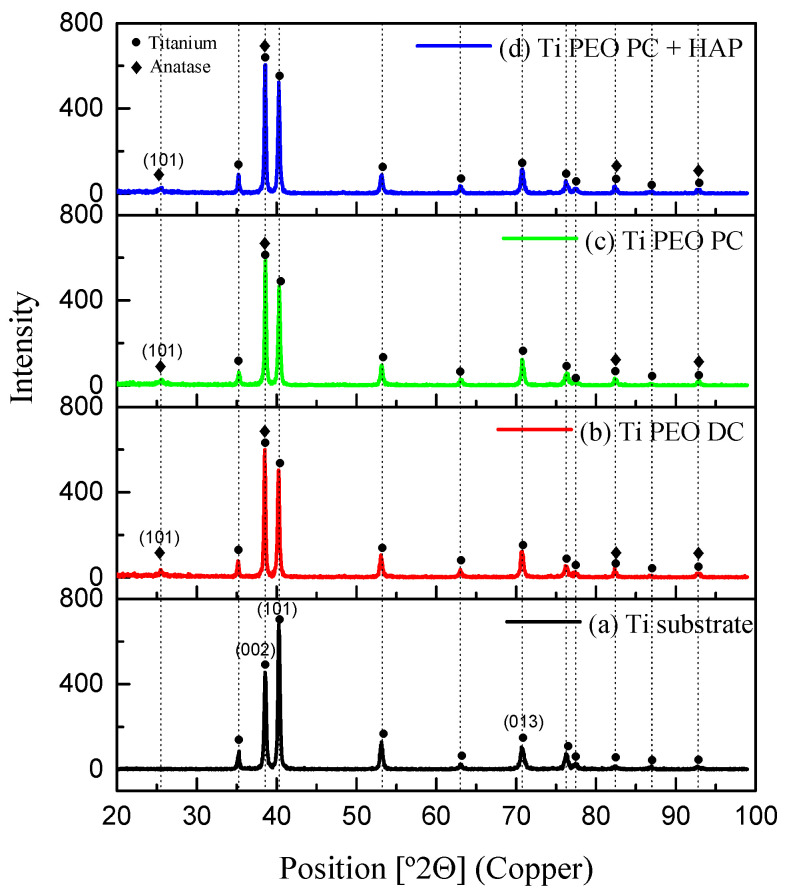
XRD diffraction patterns of Ti-based samples: (**a**) Ti substrate, (**b**) Ti PEO DC; (**c**) Ti PEO PC; (**d**) Ti PEO PC + HAP.

**Figure 6 materials-16-01410-f006:**
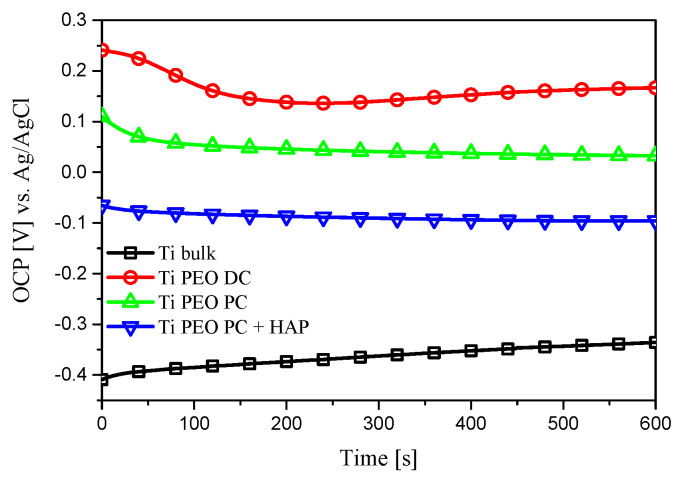
OCP measurements of the Ti-based samples for 600 s in Ringer’s solution at 37 °C.

**Figure 7 materials-16-01410-f007:**
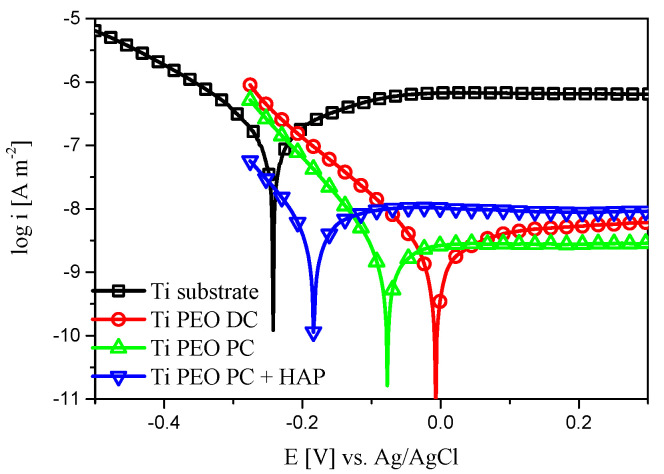
Potentiodynamic polarization curves of the Ti-based samples in Ringer’s solution at 37 °C, scan rate 0.16 mV s^−1^.

**Figure 8 materials-16-01410-f008:**
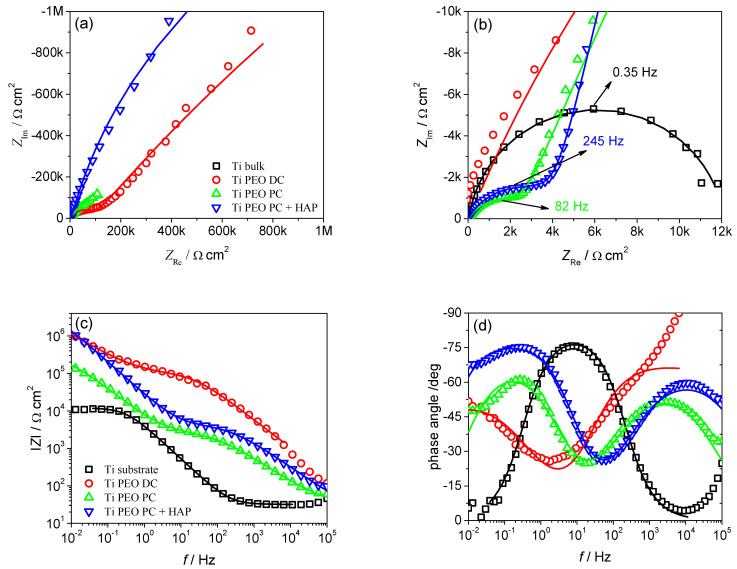
Results of experimental EIS measurements (symbols) in Ringer’s solution at 37 °C for untreated Ti and PEO-treated Ti samples: (**a**) Nyquist plots; (**b**) detail of the high-frequency part of Nyquist plots; (**c**) absolute impedance plots and (**d**) phase angle plots. Continuous lines represent the results of modeling to the EEC given in Figure 8.

**Figure 9 materials-16-01410-f009:**
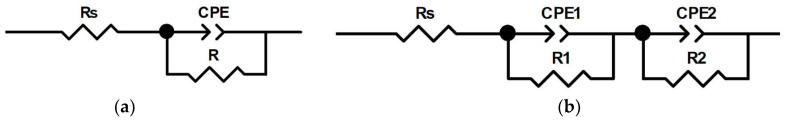
The electrical equivalent circuit was used to model the impedance data of (**a**) untreated titanium substrate and (**b**) PEO-treated titanium.

**Figure 10 materials-16-01410-f010:**
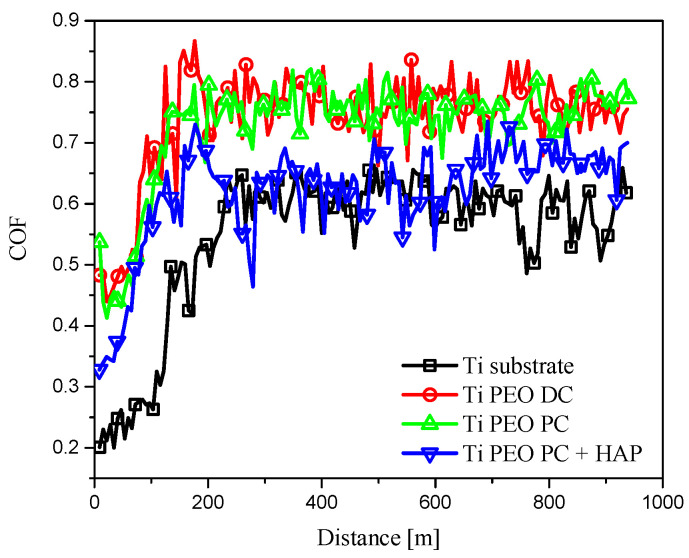
COF evolution for Ti-based samples vs. 100Cr6, *F* = 5 N, 300 rot/min.

**Figure 11 materials-16-01410-f011:**
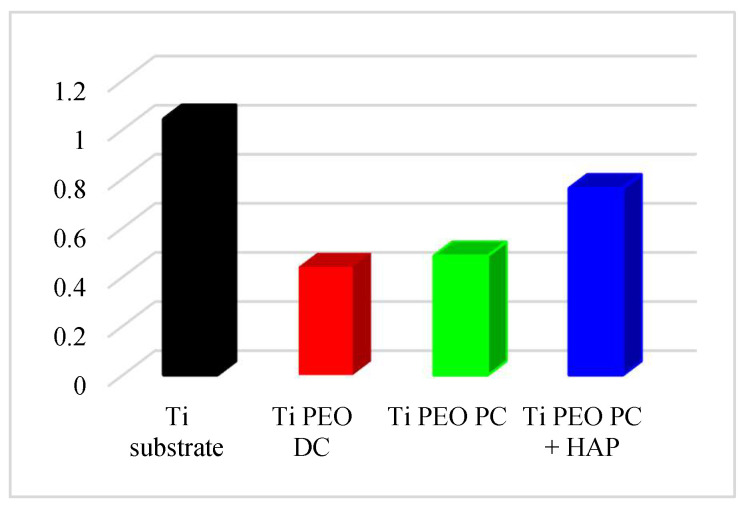
Wear rate [10^4^ mm^3^ N^−1^ m^−1^] of Ti substrate, Ti PEO DC, Ti PEO PC, and Ti PEO PC + HAP under the load of 5 N and the sliding distance of around 1000 m.

**Table 1 materials-16-01410-t001:** PEO treatment conditions for obtaining the Ti-modified samples.

Sample	Treatment Conditions				
	Current Density, *i* [mA cm^−2^]	Voltage, [V]	Total Time, [s]	*d.c.*, [%]	HAP Deposition
Ti PEO DC	10	max 240	600	-	-
Ti PEO PC			1250	20	-
Ti PEO PC + HAP			1250	20	600 s at 150 V

**Table 2 materials-16-01410-t002:** Corrosion parameters were estimated from the Tafel extrapolation method.

Sample	*i_corr_*_,_ [A cm^−2^]	*E_corr_*, [V] vs. Ag/AgCl	Corr. Rate [mm Year^−1^]
Ti substrate	687 × 10^−10^	−0.242	598 × 10^−6^
Ti PEO DC	3.30 × 10^−10^	−0.007	2.87 × 10^−6^
Ti PEO PC	3.64 × 10^−10^	−0.077	3.17 × 10^−6^
Ti PEO PC + HAP	10.1 × 10^−10^	−0.187	8.77 × 10^−6^

**Table 3 materials-16-01410-t003:** EIS parameters were obtained by fitting the experimental data of untreated Ti and PEO-treated Ti substrate.

Parameter	Ti Substrate	Ti PEO DC	Ti PEO PC	Ti PEO PC + HAP
*R*_S_/Ω	31.8 (0.7%)	20.6 (0.7%)	32.6 (1.5%)	30.6 (2.7%)
CPE-T_1_/F cm^−2^ s^n−1^	-	4.18 × 10^−7^ (3.7%)	5.20 × 10^−6^ (7.03%)	1.41 × 10^−6^ (7.3%)
*n* _1_	-	0.69 (1.4%)	0.67 (1.1%)	0.70 (1.0%)
*R*_1_/Ω cm^2^	-	11.6 × 10^4^ (6.9%)	0.25 × 10^4^ (3.1%)	0.40 × 10^4^ (2.4%)
CPE-T_2_/F cm^−2^ s^n−1^	4.35 × 10^−5^ (1.0%)	5.42 × 10^−6^ (4.9%)	3.54 × 10^−5^ (1.2%)	7.30 × 10^−6^ (1.0%)
*n* _2_	0.91 (0.2%)	0.66 (5.6%)	0.78 (1.0%)	0.86 (0.5%)
*R*_2_/Ω cm^2^	1.2 × 10^4^ (0.9%)	8.14 × 10^6^ (88%)	0.29 × 10^6^ (5.7%)	5.02 × 10^6^ (11.5%)
*χ* ^2^	2.5 × 10^−3^	1.9 × 10^−2^	4.5 × 10^−3^	5.1 × 10^−3^

## Data Availability

Not applicable.

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
