# Peer review of "Characteristics of Hydroxyapatite-Modified Coatings Based on TiO2 Obtained by Plasma Electrolytic Oxidation and Electrophoretic Deposition"

_materials, 2023, doi:10.3390/ma16041410_

Round 1

Reviewer 1 Report

1. More suitable title of the manuscript would be: “Characteristics of hydroxyapatite-modified coatings based on TiO2 obtained by plasma electrolytic oxidation and electrophoretic deposition”

2. The Abstract should not give the composition of Ringer's solution, but should indicate that PEO was carried out in a silicate-alkaline electrolyte. In the same place, the scheme for the sliding wear test is erroneously named as "pin-on-disc" instead of "ball-on-disc"

3. In the Introduction (line 41) it says "promoting galvanic or crevice corrosion". What is "galvanic corrosion"? It is known that titanium is resistant to local corrosion, including “crevice corrosion”. In reference [8] instead of "Power Metallurgy" should be "Powder Metallurgy". In reference [11] it is not about pulsed unipolar current (PC), but about alternating current (AC). Line 99 (123, 305): "additive-free suspension". What additives?

4. Materials and Methods. Figure 1 should indicate the working electrodes and counter electrodes and their polarity both in the case of PEO and in the case of electrophoresis. In table 1, one can only guess that "d.c" is "duty cycle". It would be necessary to introduce in this section the technique for measuring the thickness of coatings as their main characteristic, and then to present the results of its measurements. Line 161: "100Cr6 stainless steel ball". This is not stainless steel, but bearing steel, it contains ~1% C and ~1.5% Cr

5. Results and discussion. 3.1. Morphological characterization. The decrease in roughness for the “Ti PEO PC” sample compared to the “Ti PEO DC” sample (Fig. 3) is possibly due to a decrease in the coating thickness, since the duration of the PEO treatment is reduced from 600 s to 1250/5 = 250 s

6. Results and discussion. 3.2. EDS spectra and XRD diffraction. Figure 4 should be moved to subsection 3.2 and named "EDS spectra and XRD diffraction". Why in fig. 5 are there no peaks corresponding to silicon-containing phases? On fig. 4 silicon is a lot.

7. Results and discussion. 3.3. Corrosion behavior. Line 225: No decryption of OCP abbreviation. Line 226: "Ringer's electrolyte" should be called "Ringer's solution" (as in Abstract), and so on throughout. Line 253: "The highest absolute impedance was observed for the HAP-modified specimen." On fig. 8c, the maximum IZI is observed for the "Ti PEO DC" sample. Specimen or sample? Line 259: "Figure 9 ... CPE1-R1 corresponds to the resistance and capacitance of the porous layer." First, the other way around. Second, there is no porous layer for Figure 9a. Table 3: what is CPE-T1; n1; CPE-T2; n2?

8. Results and discussion. 3.3. Sliding wear tests. Line 279-281: "In this case, the difference between the starting COF (0.2) and the maximum reached value (0.75) denotes the poor mechanical properties…". On fig. 10 and in the Conclusions (line 318): "compared with 0.6 for the untreated titanium"?

9. Conclusions. Line 313-315: "The high corrosion resistance of the sample treated using pulsed current and HAP is confirmed also by the EIS tests, where the highest absolute impedance was observed for the HAP-modified specimens." On fig. 8c, the maximum IZI is observed for the "Ti PEO DC" sample.

10. Throughout the text and figures, "Ti bulk", "bulk Ti", "bulk titanium" should be replaced by "untreated Ti", "Ti substrate" or "bare Ti"

Author Response

Thank you very much for your valuable feedback. We took into consideration the useful suggestions and we have modified our manuscript according to your recommendations.

Attached you may find the modifications realized on the manuscript.

Reviewer 2 Report

The subject of this study is very interesting and significant from a practical point of view. Namely, the authors investigated the corrosion resistance and tribological properties of PEO-modified coatings on titanium substrates (prepared with either direct current or pulsed current).

The research is well planned and executed. However, in order to improve and clarify the whole manuscript, some parts of the work need further refinement.

-The morphology of the specimens in Figure 3 is barely visible. Please enlarge the Figure.

-Link the results of the morphological characterization to the corrosion properties of the samples. Namely, the OCP and corrosion resistance of the samples (Table 2) increase in the following order: Ti bulk < Ti PEO PC + HAP < Ti PEO PC < Ti PEO DC. The porosity and roughness of the samples increase in the same order (Figures 2 and 3). Why? Explain!!!

-Line 254: "The highest absolute impedance was observed for the HAP-modified specimen...". A similar claim appears in the conclusion. This is not evident from the results obtained (Figure 8), and according to the results in Table 3, the corrosion resistance of the samples increases in the order: Ti bulk < Ti PEO PC < Ti PEO PC + HAP < Ti PEO DC. Please correct the above information and point out the difference between the results of polarization and impedance measurements.

Author Response

(The authors gave the same response as above.)

Round 2

Reviewer 2 Report

The revised version of the manuscript has been significantly improved, and the manuscript is now available for publication in Materials.